Growth of Lahontan cutthroat trout from multiple sources re-introduced into Sagehen Creek, CA

Stead Jonathan E. 1 2
Boucher Virginia L. 3
Moyle Peter B. 1 4
Rypel Andrew L. 1 4 rypel@ucdavis.edu
1 Department of Wildlife, Fish & Conservation Biology, University of California Davis , Davis, CA , United States
2 AECOM , Oakland, California , USA
3 John Muir Institute of the Environment, University of California Davis , Davis, California , USA
4 Center for Watershed Sciences, University of California Davis , Davis , United States
Garant Dany
Electronic publication date: 2022 May 18
Publication date: 2022
Volume: 10
Electronic Location ID: e13322
Received 2021 Mar 24; Accepted 2022 Apr 1
Copyright: © 2022 Stead et al.
Copyright year: 2022
Copyright holder: Stead et al.
License: This is an open access article distributed under the terms of the Creative Commons Attribution License, which permits unrestricted use, distribution, reproduction and adaptation in any medium and for any purpose provided that it is properly attributed. For attribution, the original author(s), title, publication source (PeerJ) and either DOI or URL of the article must be cited.
License URL: https://creativecommons.org/licenses/by/4.0/

Keywords: Conservation, Broodstock management, Phenotype, Invasive species, Native species, Growth experiments, Truckee River, Negative growth, Wild trout, Fisheries management

Funding: U.S. Fish and Wildlife Service California Trout Endowment for Coldwater Fish Conservation and the California Agricultural Experimental Station of the University of California Davis CA-D-WFB-2467-H AECOM USFWS Funding was provided by the U.S. Fish and Wildlife Service. Andrew L. Rypel was supported by the Peter B. Moyle & California Trout Endowment for Coldwater Fish Conservation and the California Agricultural Experimental Station of the University of California Davis (Grant Number CA-D-WFB-2467-H). Jonathan E. Stead is employed by AECOM. USFWS was involved in the field collection of some data.

==============================
Lahontan cutthroat trout Oncorhynchus clarkii henshawi have experienced massive declines in their native range and are now a threatened species under the US Endangered Species Act. A key management goal for this species is re-establishing extirpated populations using translocations and conservation hatcheries. In California USA, two broodstocks (Pilot Peak and Independence Lake) are available for reintroduction, in addition to translocations from wild and naturalized sources. Pilot Peak and Independence Lake fish are hatchery stocks derived from native fish from the Truckee River basin and used for recovery activities in the western Geographic Management Unit Areas only, specifically within the Truckee River basin. Yet suitability of these sources for re-introduction in different ecosystem types remains an open and important topic. We conducted growth experiments using Lahontan cutthroat trout stocked into Sagehen Creek, CA, USA. Experiments evaluated both available broodstocks and a smaller sample of fish translocated representing a naturalized population of unknown origin from a nearby creek. Fish from the Independence Lake source had significantly higher growth in weight and length compared to the other sources. Further, Independence Lake fish were the only stock that gained weight on average over the duration of the experiment. Our experiments suggest fish from the Independence Lake brood stock should be considered in reintroduction efforts.

Introduction

Reintroductions and translocations are some of the few tools available for management of rare and declining species (Sarrazin & Barbault, 1996; Novak, Phelan & Weber, 2021). Need for these tools is growing rapidly, primarily as a product of the global biodiversity crisis and expansive human domination of the world’s ecosystems (Vitousek et al., 1997; Thomas, 2011; Dudgeon, 2019). Yet while potential benefits of translocations and reintroductions appear relatively straightforward, there are myriad examples of how such management efforts fail or have low effectiveness (Pérez et al., 2012; Taylor et al., 2017; Bubac et al., 2019; Robinson et al., 2020). In freshwater fisheries, one of the principal reasons for the lack of successful reintroduction and translocation efforts is the lack of scientific evaluation of various methods and approaches to facilitate management (George et al., 2009; Yackulic et al., 2021).

Lahontan cutthroat trout (Oncorhynchus clarkii henshawi) are endemic to the Lahontan Basin of northeast California, north Nevada, and south Oregon (Behnke, 1972; Behnke, 1992; Moyle, 2002; Peacock, Neville & Finger, 2018). Selection pressures on Lahontan cutthroat trout have been intense, in part because of considerable changes within the Lahontan hydrographic basin over recent geologic periods. During the mid-late Pleistocene, habitats for Lahontan cutthroat trout cycled between dendritic networks of upland streams connected with small ponds during warm and dry periods to widespread landscape inundation by Lake Lahontan–a massive endorheic lake (Madsen, Hershler & Currey, 2002; Reheis et al., 2002). At its peak, the lake likely had a surface area in excess of 12,000 km2, providing “great lake” or “inland ocean” types of habitats for Lahontan cutthroat trout, which in turn, thrived as apex predators (Madsen, Hershler & Currey, 2002). Following the Pleistocene as the landscape and climate became more arid, Lake Lahontan gradually dried into several terminal desert lakes (i.e., Pyramid and Walker Lakes) that lacked outflow, had elevated water alkalinity, and increased summer temperatures (Behnke, 1992; Reheis et al., 2002). Overall, these conditions promoted phenotypes characterized by large body sizes and rapid somatic growth. Meanwhile, populations of Lahontan cutthroat trout also persisted in smaller coldwater upland stream and oligotrophic lake habitats. Yet the lacustrine and fluvial forms of the sub-species differ from one another in distinct ways. For example, lake fish are larger, faster growing and typically have more pyloric caeca and a greater number of gill rakers–presumably because of increased piscivory (Peacock et al., 2017). The entire geographic distribution also includes two large river systems-the Quinn and Humboldt Rivers. The Quinn River was periodically inundated by the pluvial lake, but the Humboldt River was not and thus retained mainly fluvial forms.

Over the last 150 years, Lahontan cutthroat trout have vanished from the majority of their distribution due to massive stream ecosystem alterations (Griffith, 1988; Schroeter, 1998; Dunham, Cade & Terrell, 2002; Moyle, Katz & Quiñones, 2011; Peacock & Dochtermann, 2012). These loses have complicated efforts to manage the species. Translocations and reintroductions remain the primary management tool for reversing Cutthroat Trout declines in their native range (Harig, Fausch & Young, 2000; Budy et al., 2021). There are three cultivated strains (broodstocks) of Lahontan cutthroat trout available for reintroduction: Pilot Peak, Independence Lake, and a contemporary Pyramid Lake strain which is derived from Summit Lake. The Independence Lake and Pilot Peak hatchery stocks are for use in the western Geographic Management Unit (GMU) where all the lentic habitat exists. Additionally, translocations of wild and naturalized populations from streams are also used to recover cutthroat populations in ecosystems from which they were extirpated (Harig & Fausch, 2002; Peacock et al., 2010).

Integrating life-history variations (migratory vs resident; lacustrine vs fluvial) should be considered when formulating recovery strategies for species like Lahontan cutthroat trout. The Independence Lake strain (a high elevation oligotrophic lake) has largely remained intact, never having been extirpated. Lea (1968) originally recognized the importance of Independence Lake Lahontan cutthroat trout as one of few remaining self-reproducing and genetically “pure” or unhybridized populations of the subspecies. Rainbow Trout × Cutthroat Trout hybrids (Oncorhynchus mykiss × Oncorhynchus clarkii, aka ‘cutbows’) are commonly encountered. Further, Lea (1968) described how, as a neighboring watershed, Independence Lake fish were historically connected to Sagehen Creek and fish could move between areas. Genomic distinctiveness of Independence Lake and Pilot Peak fish was later confirmed (Peacock, Neville & Finger, 2018). However, as with many inland cutthroat trout populations, among-population genetic diversity is high (Peacock & Kirchoff, 2007; Peacock et al., 2017). The sole native population remaining in the Truckee River basin occurs in Independence Lake proper, Nevada County, California, and its tributary, Independence Creek (Gerstung, 1988; Peacock et al., 2017). Other populations in the basin are a product of re-establishment efforts, and wild populations are restricted to small headwater creeks isolated from non-native trout (Dunham et al., 2000; Moyle, 2002; Haak et al., 2010). It is one of only two lakes within the historic range (the other being mesotrophic Summit Lake found in the Northwest Lahontan basin GMU) to support a self-sustaining adfluvial population of Lahontan cutthroat trout (Simmons et al., 2020); however there is recent concern about hybridization in Independence Lake due to recent incursion by rainbow trout as a result of dam maintenance. An Independence Lake broodstock of Lahontan cutthroat trout is currently maintained by the California Department of Fish and Wildlife (CDFW).

The Pilot Peak strain has a different and somewhat curious history. Lahontan cutthroat trout populations in the large lakes in the Truckee River watershed (Lake Tahoe and Pyramid Lake) were completely extirpated during the 1940s (Al-Chokhachy et al., 2020). The Pilot Peak strain was discovered in a small out-of-basin stream in Utah and presumed as historic Pyramid Lake strain based on morphology; however later use of genetic data showed that these fish in fact originated from the Truckee River watershed likely either Lake Tahoe or Pyramid Lake (Peacock et al., 2017). This strain has been a major component of recent recovery efforts of the species, and a broodstock is actively maintained for use by managers by the US Fish and Wildlife Service in Gardnerville, Nevada (Al-Chokhachy et al., 2020).

Originally listed as Endangered under the US Endangered Species Act in 1970, Lahontan cutthroat trout were re-classified to Threatened in 1975, in part to facilitate increased management. Yet even with substantial efforts in recent years, most populations face a high risk of extinction over the next century due to presence of non-native trout species (Peacock & Kirchoff, 2004), degraded and fragmented habitats (Dunham, Vinyard & Rieman, 1997; Novinger & Rahel, 2003) and climate change (Moyle, Lusardi & Samuel, 2017; Muhlfeld et al., 2018). At this point, it remains unclear which of the above two broodstocks might perform best in various re-establishment efforts. Furthermore, self-sustaining stream populations of Lahontan cutthroat trout might also be available for re-establishment efforts via translocation.

The primary objective of our study was to compare early-life growth of two hatchery strains of Lahontan cutthroat trout reintroduced into Sagehen Creek, CA. Sagehen Creek was historically connected to the mainstem Truckee River prior to construction of Stampede and Boca dams. We also opportunistically evaluated a third population of naturally-reproduced fish translocated from a nearby stream. Sagehen Creek is a small headwater mountain meadow stream in the Truckee River basin which historically shared connectivity to Independence Lake (Lea, 1968). While we primarily sought to evaluate performance of two available hatchery sources, we also included a limited evaluation of a naturally reproducing population as an additional frame of reference.

Materials and Methods

Sagehen Creek is located on the eastern slope of the Sierra Nevada Mountains approximately 12 km north of Truckee, Nevada County, CA (Fig. 1) within the Sagehen Experimental Forest, where recent large-scale disturbance has been minimal. Sagehen Creek is a small, spring-fed stream originating from mountain snowmelt (~2,530 m elevation) that meanders through 10 km of forest and mountain meadow before reaching Stampede Reservoir at 1,780 m elevation. Flows in Sagehen Creek are seasonally dynamic (Seegrist & Gard, 1972); average discharge (1956–2005) = 0.35 m3 s−1, September base discharge = 0.06–0.08 m3 s−1, and peak discharge (in winter or spring) is typically higher than these values by > at least 2 orders of magnitude. Average wetted stream width in the study area during August 2006 was 3.7 m (±0.1 SE).

Figure 1 Location of Sagehen Creek, CA including study reach used in Lahontan cutthroat trout growth experiments, highlighted in pink.

Lahontan cutthroat trout disappeared from Sagehen Creek ca. 1900 coincident with extensive logging and grazing activities in the watershed. The reach where our experiment was conducted (1,950 m elevation) now supports an abundance of invertebrates, native Paiute Sculpin (Cottus beldingii), and naturalized populations of Brook Trout (Salvelinus fontinalis), Brown Trout (Salmo trutta), and Rainbow Trout (Oncorhynchus mykiss).

Two hatchery sources of Lahontan cutthroat trout were available for re-establishment: Pilot Peak and Independence Lake. Both sources derive from the Truckee River basin, and both sources are lacustrine. Fish used in the study were originally collected by the California Department of Fish and Game from Independence Lake and planted into Heenan Lake, Alpine County, California in 1975 (Somer, 2000). Reproductively mature fish in Heenan Creek (tributary to Heenan Lake) are used as a broodstock and progeny are raised in the Hot Creek Hatchery, Mono County, California USA.

Independence Lake Lahontan cutthroat trout were spawned on 01 June 2005 and raised by the California Department of Fish and Game (now the California Department of Fish and Wildlife or “CDFW”) at Hot Creek Hatchery for ~13 mo (Fig. 2). We used an undifferentiated sample of this cohort netted from hatchery raceways and transported fish to Sagehen Creek on 11 July 2006. Pilot Peak Lahontan cutthroat trout were spawned during spring 2005 by the U.S. Fish and Wildlife Service at Lahontan National Fish Hatchery in Gardnerville, Nevada. Spawning occurred from late February through April, with a peak in late March. Fish were transferred from hatchery raceways to net pens in June Lake, Mono County, California approximately 5 months later, on 01 September 2005, where they were reared for 10 additional months in accord with hatchery practices at that time. We netted an undifferentiated sample of Pilot Peak fish from pens in June Lake and transported them to Sagehen Creek on 14 July 2006. To minimize differences in body condition and to habituate fish to Sagehen Creek, both hatchery groups were held in net pens in Sagehen Creek and fed to satiation on commercial trout pellets twice daily for 1 month prior to experiment initiation.

Figure 2 Pictures of study fish and fieldwork including (A) tagging of individual fish with visual implant alphanumeric tags; (B) Lahontan cutthroat trout in Sagehen Creek; (C) Lahontan cutthroat trout in the Hot Creek Hatchery; (D) and (E) Examples of fencing and block nets used to bound experimental reaches within Sagehen Creek, CA.

We also had an opportunity for a limited evaluation of a naturalized population of naturally-reproduced Lahontan cutthroat trout from Austin Meadow Creek (~2,075 m elevation), Nevada County, California. This population was translocated by the California Department of Fish and Game from Macklin Creek (also outside the Lahontan Basin) in the early 1970s and has since developed a naturally reproducing population. This stream is a tributary of the Middle Yuba River, outside the Lahontan Basin. Fish were collected from Austin Meadow Creek using backpack electrofishing and hook and line sampling. These fish were held in net pens for 2 days prior to initiating the experiment because holding wild fish in captivity prior to release may adversely affect condition and is a technique unlikely to be employed by resource agencies in future re-introduction efforts (e.g., wild trout may not eat pellet feed). Thus, there may be experimental bias because sources were not captured and acclimated in the same way prior to initiating the experiment. However, controlling for acclimation effects would not have been possible or desirable in this study given permit constraints and the nature of differences between groups. Ultimately, the goal of the acclimation process is to treat each group in a manner that minimizes stress. For hatchery fish that go into the wild, this process often means exposing fish to the stream and hopefully getting them to feed on naturally-occurring foods (e.g., stream drift). However, fishes recruited from the wild in most cases don’t require acclimation, and there would in-practice be serious questions as to whether they would even eat pellets, which may in turn compromise survivability.

We constructed 18 temporary fish barriers enclosing nine 30 m experimental reaches over 1.7 stream km. Barriers were 1.3 cm mesh hardware cloth that enclosed the upstream and downstream sections of each 30 m reach but caused little-to-no disturbance in streamflow and drift. Mesh barriers were cleaned at least daily, sometimes more depending on debris flow conditions, such that enclosures were always clean and fully functional. To the best of our knowledge, non-natives did not recolonize experimental reaches during the study. Each experimental reach contained a deep pool, undercut banks, woody debris and rock, cobble, boulder, and gravel substrate (Fig. 2). We selected study reaches non-randomly, to control for presence of important habitat features. After completion of barriers and prior to initiation of the experiment, all non-native trout present within the study reaches were removed via multi-pass depletion using a backpack electrofisher. All native Paiute Sculpin were returned to the stream. Thus, we attempted to mimic the historical stream conditions and fish community that Lahontan cutthroat trout would have experienced historically.

Target stocking densities were 40 g m−3, representing values commonly observed from field surveys using backpack electrofishing surveys of non-native trout in Sagehen Creek (mean density = 50.3 g m−3; PB Moyle and VL Boucher, 1990–2015, unpublished data), and from backpack electrofishing surveys of Lahontan cutthroat trout in Gance and Frazer creeks NV USA (mean density = 51.4 g m−3) (Wenger et al., 2017). Densities were adjusted downwards because: (1) Lahontan cutthroat trout often exist at lower densities than non-native trout (Schroeter, 1998), (2) smaller Nevada streams provide greater visual buffering between fish than Sagehen Creek (RE Schroeter, personal communication), potentially resulting in higher densities (Chapman, 1966), and (3) the competitive advantage of larger fish due to size dominance hierarchies (Newman, 1956; Chapman, 1962) may be less pronounced at low densities (Gurevitch et al., 1992). All three sources were stocked into the same reaches, and densities of each source remained similar across reaches (Dataset S1). We measured habitat characteristics in each study reach including depth, volume, substrate size, canopy shade, density of large woody debris, and other habitat features. Further, we collected temperature data from the deepest part of each reach using a HOBO H8 logger over the full course of experiments (Onset Inc., Bourne, MA, USA, precision <0.7 °C).

Stocking of Sagehen Creek began 14 August 2006 when we haphazardly netted fish from holding pens. Immediately prior to stocking, fish were sedated, fitted with 1 × 2.5 mm medical grade elastomer alpha-numeric visual implant tags (VI tags, Northwest Marine Technologies, Inc., Shaw Island, WA, USA, Fig. 2), weighed (wet weight, ±0.05 g), and measured (standard length or “SL”, mm). One VI tag was inserted into the adipose tissue behind each eye, allowing for long-term recognition of individuals. Further, recapture of enclosed fish at the conclusion of the experiment allowed some estimation of tag efficacy (Shepard et al., 1996; Turek, Pegg & Pope, 2014), which can be higher in larger fish (Ward et al., 2015). Fish recovered ~1 h in an aerated cooler before stocking. Austin Meadow fish were stocked first, and stocking continued, alternating between hatchery sources, until addition of more fish would have resulted in a larger deviation from the target density of 40 g m−3 than would cessation of stocking. Stocking lasted 3 days total. Upon study conclusion (82 d later), fish were collected from reaches using backpack electrofishing, sedated, identified by VI tag, re-measured and -weighed, and released back to Sagehen Creek. All fish were positively identified at the end of the study based on VI tags. Mean and maximum water temperatures over all reaches ranged 6.5–6.9 °C and 0.3–15.6 °C, respectively. Growth data collected during experiments are available open access in Dataset S1.

We acknowledge growth experiments are a less common approach for managers to assess fish growth performance in streams (but see Lachance & Magnan, 1990). For example, in many situations age and growth analyses using hard parts such as otolith sagittae and/or fin rays is preferred (Fleener, 1952; Cooper, 1970), but only after assumptions of such techniques have been validated (Beamish & McFarlane, 1983) or when sacrifice of a larger number of fish is possible. In other cases, tagging and recapture of wild fishes (Myrvold & Kennedy, 2015; Uthe et al., 2016) may be desirable to gain new information on animal motility (Alexiades, Peacock & Al-Chokhachy, 2012) or to understand secondary production, i.e., accumulation of heterotrophic biomass over time (Layman & Rypel, 2020). Nonetheless, there is an important place within fisheries science for growth experiments, especially when evaluating translocation potential of sensitive populations in the absence of other information (Andrews et al., 2016). Translocation studies using caged fish are especially crucial to studying how Cutthroat Trout are limited by resources (Knight, Orme & Beauchamp, 1999; Boss & Richardson, 2002). Because Lahontan cutthroat trout have declined substantially across their range, there are many questions about how to begin successful translocation that might be addressed using experimental approaches.

We used differences in weight and length (growth) of recaptured experimental fish to evaluate performance of fish from the three broodstocks introduced into Sagehen Creek. We initially developed weight- and length-frequency histograms for fish before and after stocking for each stock (Fig. 3), and used Kolmogorov-Smirnov tests to examine whether the shape of distributions changed significantly among the two samples (periods). Comparing distributions in this manner assists with assessing any potential for length-related bias by electrofishing at the conclusion of the experiment.

Figure 3 Frequency histograms for initial and final weights and SLs of all Lahontan cutthroat trout stocked into Sagehen Creek CA.

KS Tests for SL and Weight revealed SL and weight followed the same distributions at the beginning and end of the experiments. Austin Meadow = teal blue, Independence Lake = brown, Pilot Peak = yellow.

We compared growth rates in length and weight of Lahontan cutthroat trout among sources using mixed effect models fitted with restricted maximum likelihood (REML). Mixed models are also notable for their robustness when directional assumptions of distributions are violated (Schielzeth et al., 2020). Final lengths and weights for each fish were subtracted from initial lengths and weights to estimate growth for each individual fish over the course of the experiment. Normality tests revealed that weight and length growth data were mostly normally distributed (e.g., Lilliefors Normality Tests, 5/6 Ps > 0.05). Therefore, all further analyses used raw weight and length growth data. We also note models using log-transformed or non log-transformed data yield virtually identical results. We developed two mixed effect models using growth data: one for effects of broodstock on growth in weight, and the other for growth in SL. In both models, growth in weight or SL was the dependent variable, stock was the independent variable, and reach was a random effect. Growth differences between stocks were assessed using Fisher’s Post Hoc tests. All analyses were conducted in SAS statistical software (Version 9.4, SAS Institute Inc., Cary, NC, USA) and considered significant when α < 0.05. All animal handling protocols were approved by the UC Davis Institutional Animal Care and Use Committee under protocols 11529 and 18883.

Results

We successfully recaptured 136 fish at the conclusion of the experiment, all of which retained their tags. A total of 26 fish were lost during the experiments, likely due to terrestrial or avian predators. SL and weight data followed normal distributions prior to and after the experiment (Fig. 3, KS Tests, all Ps > 0.20); thus there was little evidence of length-related bias in recapture probabilities. Growth rates of Lahontan cutthroat trout were variable under experimental conditions (Dataset S1, Table 1, Fig. 4). Across all treatment groups, negative growth in weight was common. Austin Meadow, Independence Lake and Pilot Peak fish growth ranged −7.5 to 9.8 g, −5.6 to 20 g, and −7.4 to 21 g, respectively. For weight, only Independence Lake showed a median value for growth that was positive (Fig. 4); thus on average both Austin Meadow and Pilot Peak fish lost mass over the experiment. A mixed effect model revealed weight growth of Lahontan cutthroat trout varied significantly across the three broodstocks (Mixed Effect Model; –2 Res Log(Likelihood) = 827.3, Random Effects P = 0.038; Chi-Square < 0.00001). In particular, growth of Independence Lake fish was significantly faster compered to Pilot Peak (Fisher’s Test P = 0.001) and Austin Meadow (Fisher’s Test P = 0.02). Growth did not differ between Pilot Peak and Austin Meadow (Fisher’s Test P = 0.27).

Table 1 Numbers and change in mean length and weight of Lahontan cutthroat trout stocked into (n1) and collected from (n2) each reach.

Source (n1,n2)	ΔWeight (g)	ΔSL (mm)	
Reach 1	
IL (7,5)	−2.0 ± 0.5	7.4 ± 1.0	
PP (7,7)	−5.9 ± 2.4	5.4 ± 1.5	
AM (2,2)	−3.0 ± 0.9	4.0 ± 0.0	
Reach 2	
IL (7,3)	−3.7 ± 1.0	3.3 ± 0.9	
PP (6,6)	−8.0 ± 2.2	4.0 ± 1.8	
AM (2,2)	−3.6 ± 1.7	2.0 ± 0.0	
Reach 3	
IL (8,5)	−1.7 ± 0.9	5.4 ± 1.5	
PP (8,8)	−4.3 ± 2.6	4.6 ± 1.4	
AM (2,2)	−2.3 ± 0.1	0.5 ± 0.5	
Reach 4	
IL (5,4)	1.9 ± 1.9	7.5 ± 1.6	
PP (5,4)	−2.6 ± 0.5	3.3 ± 3.0	
AM (2,1)	−7.5 ± 0.0	−1.0 ± 0.0	
Reach 5	
IL (7,4)	5.2 ± 1.1	6.8 ± 2.4	
PP (6,6)	2.8 ± 2.7	4.0 ± 1.8	
AM (3,3)	−1.0 ± 1.1	2.0 ± 2.1	
Reach 6	
IL (8,5)	2.5 ± 1.6	5.2 ± 1.0	
PP (8,7)	−4.2 ± 1.7	4.3 ± 1.6	
AM (2,0)	no data	no data	
Reach 7	
IL (10,7)	8.4 ± 1.9	8.0 ± 0.9	
PP (10,9)	5.7 ± 2.2	5.9 ± 1.5	
AM (2,2)	5.4 ± 4.5	5.0 ± 3.0	
Reach 8	
IL (12,12)	3.0 ± 1.0	7.6 ± 1.2	
PP (11,10)	−6.0 ± 2.2	−0.4 ± 1.2	
AM (3,3)	2.1 ± 3.7	1.0 ± 1.5	
Reach 9	
IL (8,7)	2.5 ± 0.7	6.6 ± 1.1	
PP (8,8)	0.7 ± 1.1	3.0 ± 1.6	
AM (3,3)	−0.7 ± 2.3	2.3 ± 1.5	
Note:

IL, Independence Lake; PP, Pilot Peak; AM, Austin Meadow. ΔWeight and ΔSL data reflect the mean ±1 SE. Raw growth data from experiments can be downloaded from Dataset S1.

Figure 4 Total growth observed for individual Lahontan cutthroat trout over the course of the experiment.

AM, Austin Meadow (teal blue circles); IL, Independence Lake (brown circles); PP, Pilot Peak (yellow circles). Horizontal black bars = median growth for each stock. Gray horizontal line denotes zero growth over the course of the experiment.

Patterns in SL growth mostly mirrored that observed for weight. However, growth in size (SL) was on average positive; all three stocks showed positive median increases in fish size (Fig. 4). Austin Meadow, Independence Lake and Pilot Peak fish growth ranged −1.5 to 8.8 mm, 2.1 to 17.3 mm, and −3.8 to 13.5 mm, respectively. A mixed effect model showed growth in length, similar to mass, differed significantly across stocks (Mixed Effect Model; –2 Res Log(Likelihood) = 747.2, Random Effects P = 0.25; Chi-Square < 0.00001). Growth of Independence Lake fish was significantly higher compered to Austin Meadow (Fisher’s Test P < 0.0001), and Pilot Peak fish (Fisher’s Test P = 0.003). Again, growth of Austin Meadow fish did not differ significantly compared to Pilot Peak (Fisher’s Test P = 0.13).

Discussion

This study is an example of the applied research needed to support conservation reintroductions and translocations. Although such tools are widely available for management of rare and declining species (Griffith et al., 1989; Olden et al., 2011), many translocation efforts fail due to a lack of scientifically-based protocols (George et al., 2009; Bubac et al., 2019). In our study, Independence Lake Lahontan cutthroat trout gained more weight and grew more in length on average compared to other evaluated stocks. In the case of weight, only Independence Lake fish showed positive growth on average under experimental conditions. We note that some individuals from all three stocks showed positive and negative growth. Furthermore, reach was a significant random effect in both models, indicating local habitat conditions are also important.

Our results parallel those from similar studies in the Sierra Nevada. In a small Sierra Nevada stream, Reimers (1963) documented a high weight loss and lack of significant growth in length for Rainbow Trout (Oncorhynchus mykiss) during the first summer and fall over five years of stocking experiments involving multiple strains. Furthermore, initial weight loss was strongly associated with high over-winter mortality of fish, especially in late winter. Increased early life growth is essential for native trout because it is often correlated with over-winter success and ability to overcome critical recruitment bottlenecks (Hunt, 1969; Coleman & Fausch, 2007). Additional studies that track reintroduced and translocated individuals for longer periods and during adult phases would be useful towards continued evaluation of various Lahontan cutthroat trout strains.

One possible explanation for observed differences in experimental growth between hatchery sources is that Independence Lake fish are pre-adapted to headwater stream environments like Sagehen Creek. This broodstock originated from Independence Lake, located <3 km north of Sagehen Creek, immediately beyond a ridge forming the northern boundary of the watershed. Prior to construction of dams and reservoirs like Stampede Reservoir downstream (Erman, 1973), Sagehen Creek and Independence Creek (outlet of Independence Lake) were adjacent tributaries of the Little Truckee River. While its name and successful brood establishment in Heenan Lake (SE of Loope CA) (Peacock et al., 2017) suggests Independence Lake fish are lacustrine-adapted fish, the degree to which these fish are truly lake-adapted is unclear. Independence Lake is actually an impoundment of Independence Creek that enlarged by ~6x a previously much smaller lake/pond from a volumetric capacity of 0.0037 to 0.0216 km3 (Berris, Hess & Bohman, 1998). Prior to construction of the dam at its outlet in 1939, the “lake” consisted of two smaller marshes that merged following impoundment. Originally, Lahontan cutthroat trout were thought to spend their entire life in only 20 m of stream (Miller, 1957); however this perspective has shifted over time, especially in more permanent and lower order rivers. In one movement study of Lahontan cutthroat trout PP strain, average distance moved in three reaches of the Truckee River ranged 0.8–1.8 km (Alexiades, Peacock & Al-Chokhachy, 2012). Further research in the Summit Lake Basin, NV demonstrated adfluvial behavior of resident Lahontan cutthroat trout and that this life history flexibility may lead to increased species persistence over time (Campbell et al., 2019). Combined these studies suggest stream size may play a role in extent to which fish move; patterns that may also have an evolutionary basis by strain.

In contrast, the Pilot Peak broodstock are native to the Truckee River basin in the western Lahontan hydrographic basin (Peacock et al., 2017). Fish in this system inhabit lacustrine habitat, but also spawn in streams and therefore exhibit both resident and migratory behaviors. It is most genetically similar to museum specimens collected from Pyramid Lake and Lake Tahoe (Peacock et al., 2017). Indeed Pyramid Lake is a remnant lake of ancient Lake Lahontan, and created near ocean-like conditions that the species once thrived in (Madsen, Hershler & Currey, 2002). Lahontan cutthroat trout in Pyramid Lake were apex predators that had large body sizes (>18.6 kg) and low niche overlap with other conspecifics (Heredia & Budy, 2018). Behnke (1992) speculated the large body size of historic Pyramid Lake fish had a genetic basis; a hypothesis that has been supported in a recent genetic study highlighting the uniqueness of this population (Peacock et al., 2017). These conditions contrast sharply with those of Independence Lake; thus it would not be surprising that strains of the species from the two locations would differ in response to reintroduction by location.

Study limitations

This experiment represents a first attempt at evaluating short-term growth differences between two broodstocks of Lahontan cutthroat trout intended for use in re-establishment efforts. Although it allowed for a realistic evaluation of short-term response to re-introduction in a headwater stream, our approach had potentially confounding issues. Due to effort required to create and maintain 18 temporary fish barriers on a daily (or more) basis, we were unable to include experimental controls, such as reaches stocked with only one source. Therefore, our results may best represent translocations in which multiple sources are used, or where trout are already present in the receiving habitat. Constriction of fish movement may have altered growth results such that fish were unable to move to obtain food resources or were “forced” into artificially higher densities that aren’t encountered any longer in extant populations. For example, in non-experimental populations, fish may be capable of behaviorally-selecting stream habitats at larger scales which could result in positive or higher growth compared to enclosed reaches. Ultimately “cage effects” are necessary by-products of experiments, and while we attempt to account for these dynamics using random effects in statistical models, they are endemic to ecological experiments involving enclosed animals (Hairston, 1989). We caution our limited experiments did not address these complicated issues and encourage interpretation in the context of our experimental design.

We note that our primary comparison was between Independence Lake and Pilot Peak hatchery-raised fish. Ultimately, we were unable to include enough naturalized Austin Meadow fish because of small sample sizes of the source population (owing to the rareness of these fish) to draw strong conclusions related to that group. Still, these results may still be valuable, therefore, we elected to include them in the study. Our results are not dissimilar from Ozer & Ashley (2013) who documented sanctuary stocks may not always be preferable to source stocks with higher heterozygosity. This study also focuses on early life growth as a measure of fitness (Roff, 1983; Rypel, 2011; Rypel, 2014); however, additional information on long-term growth and the link between growth and survivorship would be helpful (Pedersen et al., 2017). For example, our results document short-term differences in growth among stocks during late summer and early fall summer; however, growth patterns observed during other times of the year, or across years, might be different. Furthermore, the role of competition and social cues within hierarchies may also be important (Dunham & Vinyard, 1997; Knight, Orme & Beauchamp, 1999; Akbaripasand et al., 2014). So-called self-thinning is frequently documented as an important process (Vøllestad, Olsen & Forseth, 2002; Lobón-Cerviá & Mortensen, 2006; Tatara, Riley & Scheurer, 2009), and has shown specific application to Lahontan cutthroat trout (Dunham & Vinyard, 1997). Finally, we note that our results in Sagehen Creek might be different if executed in other ecosystem types, notably in more lowland and lacustrine systems, where other strains like Pilot Peak may perform best.

Conclusions

Use of the Pilot Peak broodstock is critical to recovery of Lahontan cutthroat trout populations in the large interconnected landscape of Truckee River and Pyramid Lake (Truckee River Basin Recovery Implementation Team, 2003; Al-Chokhachy et al., 2020). However, Independence Lake fish may also be useful for re-introductions in certain situations such as in small headwater streams. Given the current status of Lahontan and other Cutthroat Trout populations, translocations will continue as an essential tool for maintaining existing populations, re-establishing new populations, and ensuring preservation of sufficient genetic variation for future evolutionary change (Schultz et al., 2018). Managers should also recognize negative relationships between reproductive performance of natural populations of salmonids and proportion of hatchery fish present (Chilcote, Goodson & Falcy, 2011); thus translocations of hatchery fish should be limited to areas where the sub-species is known to have already vanished so as to prevent introgression with existing wild stocks (Yamamoto et al., 2006).

We see a high value in Sagehen Creek specifically in pioneering upland re-establishment strategies for Lahontan cutthroat trout. Sagehen Creek has a rich history as a representative Sierra coldwater stream and fishery (Needham & Jones, 1959; Seegrist & Gard, 1972; Erman, 1973; Gard & Flittner, 1974; Decker, 1989). For years, research focused on understanding interspecific and harvest dynamics of cold-water non-native trout species (Gard & Seegrist, 1972; Erman, 1986). Re-introduction of Lahontan cutthroat trout might be effective in Sagehen Creek if coupled with non-native species management. For example, installation of a weir, or a series of weirs, may assist in preventing colonization by non-native species. In addition, periodic non-native removals may be needed, perhaps following a high flow winter (or wild fire burn year with winter runoff) that would naturally drive Brook Trout populations to vulnerable levels (Seegrist & Gard, 1972; Meyers, Dobrowski & Tague, 2010). We note active fish management is a key to modern re-establishment and success of most any Lahontan cutthroat trout population.

We recognize that addition of the Austin Meadow treatment complicated the experimental design, in part because of the inability to stock enough fish to match the densities of the other strains within experimental reaches. Nonetheless, densities remained roughly equivalent by strain across the treatments such that we retain confidence in our results. Importantly, when managers are faced with the possibility of reintroducing this species to a system, they have only three options: Pilot Peak, Independence Lake or translocation of wild or naturalized fish from a nearby system. Future reintroduction studies should continue to look for differences among available broodstock strains. Building more complete understanding of the mosaic of possible management actions will be helpful for recovering Lahontan cutthroat trout populations.

Supplemental Information

Supplemental Information 1 Raw growth data for juvenile Lahontan cutthroat trout reared in experimental reaches in Sagenhen Creek, CA.

Includes the experimental reach for each fish (XR) and the unique identifier (fish_id) for every fish.

Click here for additional data file.

We thank USFWS staff for field assistance during sampling. Our study also benefited from extensive cooperation and assistance from the California Department of Fish and Game. We thank Cameron Zuber, the Moyle lab, Jim Plehn, Dan Ryan & family, Stephanie Mehalick, and others for excellent field assistance. Neil Willits at UC Davis provided valuable statistical support. Helpful reviews were provided by Richard Beamish, Sharon Lawler and the anonymous reviewers. We appreciate discussions with Patrick Crain, Joseph Cech and Gabriel Singer. Sagehen Creek Field Station at UC Berkeley provided generous facility use and equipment support.

Additional Information and Declarations

Competing Interests

Author Contributions

Animal Ethics

Data Availability

The authors declare that they have no competing interests.

Jonathan E. Stead conceived and designed the experiments, performed the experiments, analyzed the data, prepared figures and/or tables, authored or reviewed drafts of the paper, and approved the final draft.

Virginia L. Boucher performed the experiments, authored or reviewed drafts of the paper, and approved the final draft.

Peter B. Moyle conceived and designed the experiments, authored or reviewed drafts of the paper, and approved the final draft.

Andrew L. Rypel analyzed the data, prepared figures and/or tables, authored or reviewed drafts of the paper, and approved the final draft.

The following information was supplied relating to ethical approvals (i.e., approving body and any reference numbers):

UC Davis Institutional Animal Care and Use Committee (IACUC) provided full approval for this research (protocols 11529 and 18883).

The following information was supplied regarding data availability:

The raw data is available in the Supplemental File.

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
