# Peer review of "Growth of Lahontan cutthroat trout from multiple sources re-introduced into Sagehen Creek, CA"

_PeerJ, doi:10.7717/peerj.13322_

## Round 0.1 · original submission · Major Revisions

We have received two reviews for your manuscript. Both reviewers identified several important problems with the current version that deserve important revisions. Most importantly, both reviewer identified issues with the experimental design of the study (life history differences among strains used and some experimental bias).

Reviewer 1 provided important information that should be integrated to prevent the manuscript from oversimplifying the history of the study species. I concur with several points raised by reviewer 1 that suggest that the strains used could be much better described and that they have already been genetically described. Also, the study uses hatchery fish with different life histories (lacustrine and fluvial strains) which could be confounding the analyses; this needs to be addressed.

I also agree with the second reviewer who felt that the paper has a rather limited scope and represents a very specific case study. Importantly, reviewer 2 also pointed out experimental bias related to the fact that the three sources were not captured and acclimated in the same way prior to initiating the experiment. He also pointed out that the levels of intra- and interspecific competitions were not controlled for in the experimental reaches. A better discussion of all the limitations is needed.

In light of these comments, I leave an opportunity to resubmit the manuscript open. However, I strongly emphasize that all comments provided by both reviewers will need to be answered convincingly in the revised version.

·

Basic reporting

PeerJ review
Growth of Lahontan cutthroat trout from multiple sources re-introduced into Sagehen Creek, CA (#59331)
The cutthroat trout of the western United States face an uncertain future due to the impacts of habitat fragmentation and degradation, the introduction of non-native salmonids and climate change. Lahontan cutthroat trout have been extirpated from >90 % of their historic fluvial habitat and 99% of their historic lacustrine habitat. The authors of this paper do not convey to the reader the rich history of this subspecies, and therefore the significance of their results is hard to judge given the lack of a larger context. They also do not do justice to the life history variation and challenges of recovery of this ESA listed subspecies. In my opinion, the introduction should be completely rewritten, to include some of the hydrological and geological history of the Lahontan hydrographic basin where this fish is found. The pluvial Lake Lahontan of the Pleistocene played a huge role in the evolution of this trout. The large Humboldt River watershed in the eastern portion of Nevada was never inundated by the large lake, which covered most of northwestern Nevada 10,000 years ago. The remaining lacustrine habitat is found in the western portion of the range in the Truckee and Walker watersheds. The lacustrine and fluvial forms of this fish differ in growth and phenotype (lake fish have more pyloric caeca and greater number of gill rakers) and they use both life history variants in this study. The authors give no history of the hatchery stocks they use in this study. The Independence Lake strain (a high elevation oligotrophic lake) has remained intact never having been extirpated. The Pilot Peak strain has a very interesting history. The Lahontan cutthroat trout populations in the large lakes in the Truckee River watershed- Lake Tahoe (oligotrophic) and Pyramid Lake (mesotrophic and endorheic) were extirpated in the 1940s. The Pilot Peak strain (see Peacock et al. 2017) was discovered in a small out-of-basin stream in Utah and identified as a putative historic Pyramid Lake strain by its morphology and then using genetic data shown to have come from the Truckee River watershed likely either Lake Tahoe or Pyramid Lake. This information should be included to set the stage for their study.

Line 28 – this population (Macklin Creek) was planted in these waters sometime in the 20th century and so they are not representative of a population of wild Lahontan cutthroat trout.
Lines 38-39 Include a reference pluvial Lahontan as the fluvial habitat of this fish (e.g., Quinn and Humboldt) is not alkaline and never was. Should not refer to the Great Basin but rather the Lahontan hydrographic basin as Bonneville cutthroat are also in the Great Basin. Should make distinction between fluvial and lacustrine forms. Also this fish was found in oligotrophic habitat as well as alkaline habitat in the same watershed (Truckee River).
The differences among these strains and they are different genetically – may explain their results. Growth rate differs among strains of Lahontan cutthroat trout and in different habitats and among life histories – fluvial vs lacustrine.
Lines 54-55 over simplification and wrong citation –Peacock and Kirchoff 2004 has nothing to do with defining distinct population segments
Line 66 you have to talk about this strain more in depth with its unique history
Line 67 this is vague
Line 95 oversimplification – lack of context
Lines 122-124 these different lineages have been genetically characterized (see Peacock and Kirchoff 2007; Peacock et al. 2017).
Lines 142-144 Where are all of the more current references for this species? Where are Jason Dunham's papers?
Lines 175-176 such as?
Lines 177-178 condition factor? Rates of growth among these fishes may be genetically determined and differ among strains.
Lines 271 need citation here - and Pilot Peak derives from the Truckee River basin - evidence for Lake Tahoe origin as well.
Lines 279-281 how do short term growth rates really inform recovery?

Experimental design

The authors use hatchery fish with different histories and different life histories. This is problematic and the authors need to be clearer about these differences and potential confounding factors.

Validity of the findings

Hard to say much about translocation success with results of 3 months of an in stream experiment. There is so much information left out that for anyone with knowledge of this cutthroat trout species there are many questions. They use both lacustrine and fluvial strains in this study and that in itself is a red flag

Additional comments

PeerJ review
Growth of Lahontan cutthroat trout from multiple sources re-introduced into Sagehen Creek, CA (#59331)
The cutthroat trout of the western United States face an uncertain future due to the impacts of habitat fragmentation and degradation, the introduction of non-native salmonids and climate change. Lahontan cutthroat trout have been extirpated from >90 % of their historic fluvial habitat and 99% of their historic lacustrine habitat. The authors of this paper do not convey to the reader the rich history of this subspecies, and therefore the significance of their results is hard to judge given the lack of a larger context. They also do not do justice to the life history variation and challenges of recovery of this ESA listed subspecies. In my opinion, the introduction should be completely rewritten, to include some of the hydrological and geological history of the Lahontan hydrographic basin where this fish is found. The pluvial Lake Lahontan of the Pleistocene played a huge role in the evolution of this trout. The large Humboldt River watershed in the eastern portion of Nevada was never inundated by the large lake, which covered most of northwestern Nevada 10,000 years ago. The remaining lacustrine habitat is found in the western portion of the range in the Truckee and Walker watersheds. The lacustrine and fluvial forms of this fish differ in growth and phenotype (lake fish have more pyloric caeca and greater number of gill rakers) and they use both life history variants in this study. The authors give no history of the hatchery stocks they use in this study. The Independence Lake strain (a high elevation oligotrophic lake) has remained intact never having been extirpated. The Pilot Peak strain has a very interesting history. The Lahontan cutthroat trout populations in the large lakes in the Truckee River watershed- Lake Tahoe (oligotrophic) and Pyramid Lake (mesotrophic and endorheic) were extirpated in the 1940s. The Pilot Peak strain (see Peacock et al. 2017) was discovered in a small out-of-basin stream in Utah and identified as a putative historic Pyramid Lake strain by its morphology and then using genetic data shown to have come from the Truckee River watershed likely either Lake Tahoe or Pyramid Lake. This information should be included to set the stage for their study.

Line 28 – this population (Macklin Creek) was planted in these waters sometime in the 20th century and so they are not representative of a population of wild Lahontan cutthroat trout.
Lines 38-39 Include a reference pluvial Lahontan as the fluvial habitat of this fish (e.g., Quinn and Humboldt) is not alkaline and never was. Should not refer to the Great Basin but rather the Lahontan hydrographic basin as Bonneville cutthroat are also in the Great Basin. Should make distinction between fluvial and lacustrine forms. Also this fish was found in oligotrophic habitat as well as alkaline habitat in the same watershed (Truckee River).
The differences among these strains and they are different genetically – may explain their results. Growth rate differs among strains of Lahontan cutthroat trout and in different habitats and among life histories – fluvial vs lacustrine.
Lines 54-55 over simplification and wrong citation –Peacock and Kirchoff 2004 has nothing to do with defining distinct population segments
Line 66 you have to talk about this strain more in depth with its unique history
Line 67 this is vague
Line 95 oversimplification – lack of context
Lines 122-124 these different lineages have been genetically characterized (see Peacock and Kirchoff 2007; Peacock et al. 2017).
Lines 142-144 Where are all of the more current references for this species? Where are Jason Dunham's papers?
Lines 175-176 such as?
Lines 177-178 condition factor? Rates of growth among these fishes may be genetically determined and differ among strains.
Lines 271 need citation here - and Pilot Peak derives from the Truckee River basin - evidence for Lake Tahoe origin as well.
Lines 279-281 how do short term growth rates really inform recovery?

·

Basic reporting

The manuscript meets most aspects listed in the guidelines provided by the journal about writing, structure, figures, tables, raw data shared, self-contained with relevant results to hypotheses (objectives in this case), although some aspects would merit clarifications (see my specific comments below).

Experimental design

The research question is well defined, relevant and meaningful, and identify how the study fills an identified knowledge gap, but is of a limited scope.

The fact that fish density stocked in experimental reaches differed among strains is a major shortcoming because the levels of intra- and interspecific competition were not controlled in the experiment (see my detailed comment below).

Validity of the findings

The impact and novelty of the study are not assessed as well as the rationale and benefit to literature.

The results of the study are difficult to interpret due to the fact that fish density stocked in experimental reaches differed among strains (see my detailed comment below).

Additional comments

The objective of this study was to compare early-life growth of two hatchery and one wild sources of Lahontan Cutthroat Trout reintroduced into Sagehen Creek, CA, in order to evaluate their potential for re-introduction efforts into small montane headwater streams in California. Lahontan Cutthroat Trout have experienced massive declines in their native range and are now a threatened species under the US Endangered Species Act.

The study provides some interesting results but I have concerns regarding the experimental manipulations and design:

1- Lines 102-120: Fish from the Pilot Peak and Independence Lake sources were netted from hatchery raceways, held in net pens in Sagehen Creek and fed to satiation on commercial trout pellets twice daily for one month prior to experiment initiation. Those from the wild source were collected by electrofishing and hook and line sampling from Austin Meadow Creek, held in net pens for two days prior to initiating the experiment (“because holding wild fish in captivity prior to release may adversely affect their condition”. There is an experimental bias here because the three sources were not captured and acclimated in the same way prior to initiating the experiment. The stress and short acclimation experienced by the wild source should be an issue discussed in the manuscript.

2- Although this was the case, it is not explicitly mentioned in the Methods section that the three sources were stocked into the same reaches.

3- Lines 293-295 and Table 1: The authors were unable to include enough wild Austin Meadow fish because of small sample sizes of the source. This is a major shortcoming because the levels of intra- and interspecific competitions were not controlled in the experimental reaches. It would have been preferable to adjust the density of the Pilot Peak and Independence Lake sources to that of the Wild Austin Meadow to control for these effects (and may be limit the study to fewer reaches).

4- Lines 126-128: I am surprised that a 1.3 cm mesh hardware cloth that enclosed the upstream and downstream sections of each reach caused little-to-no disturbance in streamflow and drift. From my experience, even a 1.3 cm mesh hardware cloth should clog relatively rapidly with debris from the drift (matter of hours or few days, depending on the stream). More details should be given regarding the mention “Due to effort required to create and maintain 18 temporary fish barriers on a daily basis…” (lines 282-283).

5- Lines 145-147: The authors mention that they measured habitat characteristics in each study reach including depth, volume, substrate size, canopy shade, density of large woody debris, ….and other habitat features (???)….and temperature data. What they did with these data? All data collection described in a Methods section should be referred to in the Results and Discussion sections.

6- Lines 154-155: The authors mention “recapture of enclosed fish at the conclusion of the experiment allowed some estimation of tag efficacy”. Again, if presented in the Methods section, this should be referred to in the Results and Discussion sections.

7- Lines 131-134: “all non-native trout present within the study reaches were removed via multi-pass depletion using a backpack electrofisher; however all native Paiute Sculpin were returned to the stream”. Why only Paiute Sculpin were returned to the stream? What did the authors do with the non-native trout? This should be specified.

8- Lines 166-167: “We acknowledge growth experiments are a less common approach for managers to assess fish growth performance”. May be in stream but not in lake ecosystems. Many works have documented growth performance of different strains of salmonids after stocking between 1960 and 1990. See for ex. Lachance & Magnan (1990) and references therein.

9- Lines 181-183: It is not clear to me how comparing the gain (or lost) in length and weight distributions during the experiment allow to assess for potential length-related bias in recapture probabilities.

10- Lines 185-187: “Mixed effect models are increasingly applied to complex fisheries datasets where pseudoreplication and uneven data structures are common”. This is now well recognized. Not necessary.

11- Lines 198-201: “Whereas effects of habitat were controlled (to the extent possible) by creating analogous experimental stream reaches, we do not present models that analyze the effect of habitat on growth. However, we did build such models and found they differed little from above; therefore we present simple models for better accessibility”. It would have been preferable to present these models and let the readers judge. The authors should have presented all the models and select the best with a procedure like the Akaike Information Criterion for small sample size (Burnham & Anderson, 2002).

12- Lines 210-211: The fact that negative growth in weight was common across all treatment groups suggest that the resources were either depleted or insufficient to maintain positive gain in weight at this fish density, and the authors have recognized this in the discussion (lines 286-288). This complicates the interpretation of the data together with the fact that the fish density differed among strains (re. comment #3 above).

13- Lines 240-241 and: “Independence Lake brood stock may be useful for re-introduction efforts into small montane headwater streams in California like Sagehen Creek”. “may be useful”; this conclusion is rather vague.

14- In the last part of the discussion, the authors confirm that the scope of their study is rather limited (to Sagehen Creek): lines 306-307: “Finally, we note that our results in Sagehen Creek might simply be different if executed in other streams and ecosystem types”; lines 324-325 “We see a high value in Sagehen Creek specifically in pioneering upland re-establishment strategies for Lahontan Cutthroat Trout”; and lines 329-330: “Re-introduction of Lahontan Cutthroat Trout might be effective in Sagehen Creek if coupled with non-native species management”.

In conclusion, this study required important field efforts and could be of interest for a local audience due to its limited scope. However, it would need an important revision, especially a better discussion of the study's limitations in relation to my main concerns (because nothing can be done at this point to improve the study design). On the other hand, it is clearly a case study and I do not know the PeerJ standards regarding the impact and contribution of manuscripts accepted for publication.

References
Burnham, K. P., & Anderson, D. R. (2002). Model selection and multimodel inference: A practical information-theoretic approach. Springer-Verlag, New York.

Lachance, S., & Magnan, P. (1990). Performance of domestic, hybrid, and wild strains of brook trout, Salvelinus fontinalis, after stocking: the impact of intra- and interspecific competition. Canadian Journal of Fisheries and Aquatic Sciences, 47, 2278–2284.

---

## Round 0.2 · Major Revisions

We managed to obtain reviews from previous reviewers on your manuscript. While both found your new version to be much improved, they also identified numerous outstanding problems with the current manuscript. Most of their concerns are minor, but since there are several of them, I decided to give a major revisions decision. I am convinced that if you thoroughly follow their additional suggestions, you will further improve your manuscript.

·

Basic reporting

see attached review

Experimental design

see attached review

Validity of the findings

see attached review

Additional comments

see attached review

·

Basic reporting

NOTE : All comments of my review of the revised version will be presented in this section

I have read the responses to the referees' concerns as well as the revised version of the manuscript and I consider that the authors have done a thorough job in revising their manuscript. Most of my concerns have been addressed and the manuscript revised accordingly. Therefore, I recommend that the manuscript be accepted with the following minor revisions:

MAIN POINT
One of my main concerns of the original version was that the study is limited in scope and falls in the category of case studies. The authors have offered a response to this concern in the last part of their rebuttal, which I do not consider satisfactory.

My suggestion would be to start the introduction with a general paragraph that would put the study in a broader context. To this respect, the first very good sentence of the discussion should be moved to the beginning of the introduction and used as a starting point to develop this first paragraph (without reference to the Lahontan Cutthroat Trout).

Then, at the beginning of the discussion, the authors could bridge to this first paragraph of the introduction by mentioning that their study is a good example of the research that should be done to better support/frame conservation reintroductions.

MINOR POINTS

NOTE: Line number refers to the track changes version

Lines 193-195: The explanation offered by the authors in the rebuttal is clearer and more explicit than this sentence of the manuscript: “Thus while acclimation techniques differed between the two primary treatment groups and the Austin Meadow fish, the goal was the same – to treat each group in a manner that minimizes stress.” Please use the explanation presented in the rebuttal to clarify the manuscript.

Lines 221-222: Not clear. Why “but” instead of “and” in this sentence: “All three sources were stocked in the same reaches, "but" densities of each source remained similar across the reaches (Supplementary Dataset 1).” Please clarify.

Lines 322-325: Delete these two sentences. They are not necessary and repeat some aspects of the introduction. Furthermore, the use of “case study” in not necessary through the manuscript. The discussion should start with the text of line 325 “In our study, Independence Lake Lahontan Cutthroat Trout gained more weight and…”, given the suggested change in my main point, above.

Lines 456-458: Delete theses three sentences. This is not the kind of considerations that we find in a scientific paper. The paragraph should start with “We recognize that addition of the Austin Meadow treatment complicated the experimental design, ….”

Experimental design

See my point #1 above

Validity of the findings

See my point #1 above

Additional comments

See my point #1 above

---

## Round 0.3 · accepted · Accept

I am satisfied with the final revisions performed on the manuscript.